# Risk factors for postoperative ileus in hysterectomy: A systematic review and meta-analysis

Zhuoer Hou[1][☯], Ting Liu[2][☯], Xiaoyan Li[1], Hangpeng Lv[3], Qiuhua Sun [1] *

1 The School of Nursing, Zhejiang Chinese Medical University, Hangzhou, China, 2 The School of Basic Medicine, Zhejiang Chinese Medical University, Hangzhou, China, 3 Baotou Medical College, Inner Mongolia Medical University, Hohhot, China

☯ These authors contributed equally to this work.
* sunqiuhua@zcmu.edu.cn

**Data Availability Statement:** All data are in the manuscript and/or Supporting information files.

**Funding:** This project is jointly supported by The National Natural Science Foundation of China (No. 8197152282). The funders had no role in study

## Abstract

### Objective

The study intended to evaluate the risk factors of postoperative ileus in hysterectomy patients.

### Study design

Systematic review and meta-analysis.

### Methods

This study conducted a systematic review and meta-analysis in accordance with the Preferred Reporting Program for Systematic Review and Meta-analysis statement. PubMed, Web of Science, Embase, the Cochrane Library and China National Knowledge Internet were searched. The search period was restricted from the earliest records to March 2024. Key words used were: (hysterectomy) AND (postoperative ileus OR postoperative intestinal obstruction OR ileus OR intestinal obstruction). Two researchers screened literatures and extracted data, and used Newcastle-Ottawa scale and Joanna Briggs Institute critical appraisal checklist for analytical cross-sectional studies to evaluate their quality. Then, Stata17 software was used for statistical analysis.

### Result

A total of 11 literatures were included. Personal factors and previous history of disease factors of postoperative ileus in hysterectomy patients included use opioids (OR = 3.91, 95% CI: 1.08–14.24), dysmenorrhea (OR = 2.51, 95%: 1.25–5.05), smoking (OR = 1.55, 95%: 1.18–2.02), prior abdominal or pelvic surgery (OR = 1.46, 95%CI: 1.16–1.83) and age (OR = 1.03, 95%: 1.02–1.04). Surgery-related factors included perioperative transfusion (OR = 4.50, 95%CI: 3.29–6.16), concomitant bowel surgery (OR = 3.79, 95%CI: 1.86–7.71), anesthesia technique (general anesthesia) (OR = 2.73, 95%CI: 1.60, 4.66), adhesiolysis (OR = 1.97, 95%CI: 1.52–2.56), duration of operation (OR = 1.78, 95%CI: 1.32–2.40), operation

design, data collection and analysis, decision to publish, or preparation of the manuscript.

**Competing interests:** The authors have declared that no competing interests exist.

approach (laparoscopic hysterectomy) (OR = 0.43, 95%CI: 0.29–0.64) and operation approach (vaginal hysterectomy) (OR = 0.35, 95%CI: 0.18–0.69).

## Conclusions

The results of this study were personal factors and previous history of disease factors, surgery-related factors, which may increase the risk of postoperative ileus in hysterectomy patients. After the conclusion of risk factors, more accurate screening and identification of high-risk groups can be conducted and timely preventive measures can be taken to reduce the incidence of postoperative ileus.

## Trial registration

The study protocol for this meta-analysis was registered (CRD42023407167) with the PROSPERO database (www.crd.york.ac.uk/prospero).

## Introduction

Postoperative ileus (POI) refers to the temporary interruption of postoperative gastrointestinal dynamics. Its pathogenesis includes over-activation of sympathetic nervous system, intestinal inflammatory response and surgical trauma, etc. [1]. The main symptoms are oral feeding intolerance, nausea, vomiting, abdominal pain and distension, delayed defecation and flatus, etc. [2]. POI is most common in abdominal surgery, especially those involving direct operation of the intestine, which has been extensively researched [3]. In abdominal surgery, benign disease hysterectomy is the most common gynecological surgery in the world. POI occurs in about 2% of patients after hysterectomy and is one of the most severe subtypes of POI [4]. Statistics show that more than 600,000 hysterectomies are performed every year [5]. 33% of women in the United States have had a hysterectomy before the age of 60 [5]. Therefore, there may be thousands of people who may develop POI after hysterectomy [6–8]. When this happens, increasing the patient's hospital stay by an average of 3.7 days, and estimated medical costs for POI range from $750 million to $1 billion per year, undoubtedly increasing the burden on medical care and families [9]. Therefore, understanding related risk factors is of guiding value for understanding POI and accelerating surgical rehabilitation of patients after hysterectomy. Not only that, it can help to develop targeted interventions to reduce the incidence of POI and reduce the medical and family burden. At present, there are some controversies on the risk factors of POI at home and abroad, the pathogenesis of POI and the effective preventive measures are not clear. Therefore, this study conducted a meta-analysis of the risk factors of POI in patients undergoing hysterectomy, summarized the risk factors, and promptly prevented the occurrence of POI and took intervention measures.

## Materials and methods

This systematic review was registered in the International Prospective Register of Systematic Reviews (CRD42023407167). The review protocol of this study could also be accessed on it. And this article was conducted under the guidance of the Preferred Reporting Program for Systematic Review and Meta-analysis, Cochrane Meta-analysis and systematic reviews and the Meta-analysis of Observational Studies in Epidemiology guidelines [10–12].

## Search strategy

The authors conducted a comprehensive literature search using PubMed, Web of Science, Embase, the Cochrane Library and China National Knowledge Internet. The search period was restricted from the earliest records to March 2024. Exact search terms used on each database (and each platform) are provided in S1 Table. Databases were searched separately, as opposed to multiple databases being searched simultaneously on the same platform. The search results of electronic data base were exported to Endnote X9 software. Results were deduplicated using Endnote software. Two authors (ZH and TL) screened titles and abstracts generated by the search independently and then assessed the full texts of all relevant articles against the inclusion criteria. In case of disagreements, a senior author (QS) was consulted.

## Eligibility criteria

The inclusion criteria were as follows:

1. The subjects were patients over 18 years of age who underwent elective hysterectomy.

2. The contents of articles included the influencing factors and risk factors of POI in hysterectomy, and the results were converted to odds ratio (OR) values, 95% confidence interval (CI) and standard error (SE).

3. Demonstrate clear criteria for POI diagnosis, and at least one of the following diagnostic criteria is met: ①Feeding intolerance, nausea, vomiting, abdominal pain and bloating, delayed defecation and flatulence 2 days or more after surgery; ②Radiography confirms the diagnosis of POI.

4. The study design should be observational study.

   The exclusion criteria were as follows:

1. Repeated publication or incomplete literature.

2. The data extraction could not be carried out or the data extraction did not meet the research requirements.

3. Literature review, systematic review, animal experiments, case reports.

4. The POI have been the result of an intra-abdominal complication.

## Definitions of outcome measures

We used the definition of POI from the previous original study, which consisted of both symptomatic evaluation and radiologic diagnosis, as described in the previous section on inclusion criteria. Additionally, risk factors for POI were defined as factors identified or suspected to increase the risk of POI during the patient's perioperative period. These variables need to be significant, as well as at least be considered risk factors in the bivariate analysis. These factors could be secondary outcomes in the literature included in this study, not necessarily primary outcomes.

## Data extraction

Pre-designed data extraction table was used to extract data independently and cross-check. The data included the following variables: first author, year of publication, country, study

design, surgical type, time to POI, sample, POI patients, risk factor. Discrepancies were resolved by consensus.

### Risk of bias and quality assessment in included studies

Newcastle-Ottawa scale (NOS) was used to evaluate the quality of the included literature, which included two scales, case-control study and cohort study [13]. The two scales each contained 3 dimensions and a total of 8 items. The full score of the two scales were 9, and a score of 4–6 for medium quality literature and a score of 7–9 for high quality literature were included in the study [13]. For cross-sectional studies, we used Joanna Briggs Institute (JBI) critical appraisal checklist for analytical cross-sectional studies [14]. The list has eight evaluation items, each corresponding to four answers: not applicable, unclear, no or yes. The higher the number of "yes" answers, the higher the quality of the study.

### Data synthesis and statistical analysis

The data were analyzed by Stata17 software, and the OR values of the original data in the literature and 95%CI were used as the combined effect size. $I^2$ value was used to analyze the heterogeneity. If $P > 0.1$ and $I^2 \leq 50\%$, there was no statistical heterogeneity among the studies, the fixed-effects model was used to analyze the heterogeneity. If $P \leq 0.1$ and $I^2 > 50\%$, the heterogeneity among studies was high, the random-effects model was used for analysis. The pooled estimates were presented with forest plots.

### Sensitivity analysis

Sensitivity analysis was conducted by using the method of article-by-article exclusion to observe whether there was a significant change in heterogeneity after the exclusion of references, and to observe the change of the OR value. If heterogeneity changed significantly after the exclusion of each paper, the paper might be the source of heterogeneity, and analyze why it became the source of heterogeneity. Alternatively, sensitivity analysis could be performed by changing the merge model. If the combined results of fixed-effects model and random-effects model are inconsistent, it meant instability.

The funnel plot was used to analyze publication bias for variables with more than 10 included literatures. If the funnel plot was symmetrical on both sides, it indicated that the possibility of publication bias was small; otherwise, it indicated that the possibility of publication bias was large. $P < 0.05$ was considered statistically significant.

## Results

### Identification and characteristics of included studies

We identified 1644 articles in the initial search of the literature and additional records through other sources. Then, by removing duplicates, screening title abstract and full text, we finally included 11 articles [7, 15–24]. The detailed selection process is presented in Fig 1. Most of the analyzed studies were retrospective cohort studies (n = 7). Most of the studies were conducted in USA (n = 5), 3 in China, the others are in France, Canada and Japan, respectively. A total of 59808 patients who underwent hysterectomy from 1994 to 2023 were included in these studies, of whom 758 developed POI (1.3%). Two of the studies included cases of hysterectomy with malignant and benign indications, seven studies with benign indications of hysterectomy and two studies with malignant indications of hysterectomy.

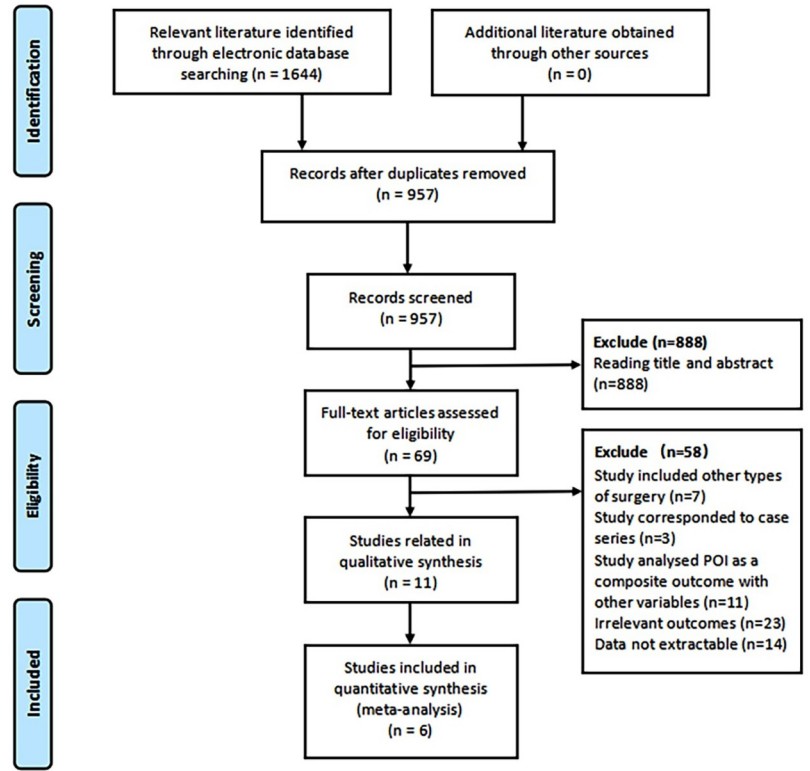

**Fig 1. PRISMA flow diagram for included studies.**

## Meta-analysis of the risk factors for POI

**Characteristics of the included population.** The six studies included in meta-analysis were all evaluated for hysterectomy [7, 15–19]. POI was defined differently among the studies included in the meta-analysis. Five studies defined POI as inability to tolerate oral administration for at least 2 days after surgical intervention, accompanied by nausea, vomiting, abdominal pain, and abdominal distention [7, 15–17, 19]. One study did not clearly identify the specific number of days to diagnose POI, relying instead on and radiological characteristics [18] (Table 1).

**Bias assessment of the included studies.** The NOS was used for the 7 cohort studies and 2 case-control studies. The articles included all scored 7–9, which is considered moderate to high quality. The JBI critical appraisal checklist was used to assess the quality of the 2 cross-sectional studies. Both studies satisfy all items. The quality assessment is shown in Figs 2–4, respectively.

## Risk factors for POI

There are twelve influenced factors for POI in hysterectomy patients included in this study. They can be classified as personal factors and previous history of disease and surgery-related factors. The heterogeneity test and meta-analysis results of each influencing factor were shown below.

**Personal factors and previous history of disease.** By analyzing the included articles, five personal factors and history of disease factors were assessed, including use opioids,

**Table 1. Summary of the studies assessed in the meta-analysis.**

| Study | Country | Study design[a] | Surgery type | Pathological | Diagnosis of POI | Study sample | POI patients[b] | Risk factors[c] |
|---|---|---|---|---|---|---|---|---|
| Fu, 2023 [15] | China | CS | Hysterectomy | Benign | 2 days | 88 | 11 | 5,8,22 |
| Gu, 2021 [16] | China | CS | Hysterectomy | Benign | 2 days | 200 | 25 | 5,8,22 |
| Arabkhazaeli, 2020 [18] | USA | RC | Hysterectomy | Benign/ Malignant | Radiographically/ Surgically | 1630 | 40 | 1,2,3,5,12,13,16,19,20 |
| Li, 2020 [19] | China | RC | Hysterectomy | Benign | >2days | 1017 | 94 | 4,5,6,7,8 |
| Sheyn, 2019 [7] | USA | RC | Hysterectomy | Benign | 3 days | 47937 | 286 | 1,5,6,9,10,11,12,13,14,16,17,19,20,21,25,26,27,28,29 |
| Antosh, 2013 [17] | USA | CC | Hysterectomy | Benign | 2–4 days | 432 | 144 | 4,5,6,10,11,12,13,16,17,18,19,21,30 |
| Muffly, 2012 [23] | USA | RC | Hysterectomy | Benign | 1 day, Radiographically | 3229 | 17 | 1 |
| Schindlbeck, 2008 [21] | Germany | RC | Hysterectomy | Benign/ Malignant | No report | 233 | 2 | 1 |
| Al-Sunaidi, 2006 [20] | Canada | RC | Hysterectomy | Benign | Radiographically | 4712 | 100 | 1,5 |
| Fujita, 2005 [22] | Japan | CC | Hysterectomy | Malignant | 1 week | 217 | 28 | 24 |
| Montz, 1994 [24] | USA | RC | Hysterectomy | Malignant | No report | 113 | 11 | 5,23 |

[a] RC: retrospective cohort, CC: case–control study, CS: cross-sectional study

[b] Criteria for POI diagnosis: ①Feeding intolerance, nausea, vomiting, abdominal pain and bloating, delayed defecation and flatulence 2 days or more after surgery; ②Radiographically or surgically. Meet one of the above criteria.

[c] Risk factors: 1.Operation approach (laparoscopic/vaginal hysterectomy); 2.Bowel injury; 3.Malignant uterine pathology; 4.Anesthesia technique (general anesthesia); 5. Adhesiolysis; 6.Duration of operation; 7.Previous cancer; 8.Dysmenorrhea; 9.Wound size > 3; 10.Perioperative transfusion; 11.Nonwhite race; 12.Age; 13.Prior abdominal or pelvic surgery; 14.Uterine weight > 250g; 15.No IMRT (Intensity Modulated Radiation Therapy); 16.Body mass index (BMI); 17.Concomitant bowel surgery; 18.Perioperative complications: cystotomy; 19.Smoking; 20.Diabetes; 21.Endometriosis; 22.Use opioids; 23.Concomitant radiotherapy; 24.Paraaortic lymph node dissection (PAND); 25.Hypertension; 26.Dyspnea; 27.Chronic obstructive pulmonary disease (COPD); 28.Coagulopathy; 29.Steroid use; 30.Postmenopausal status.

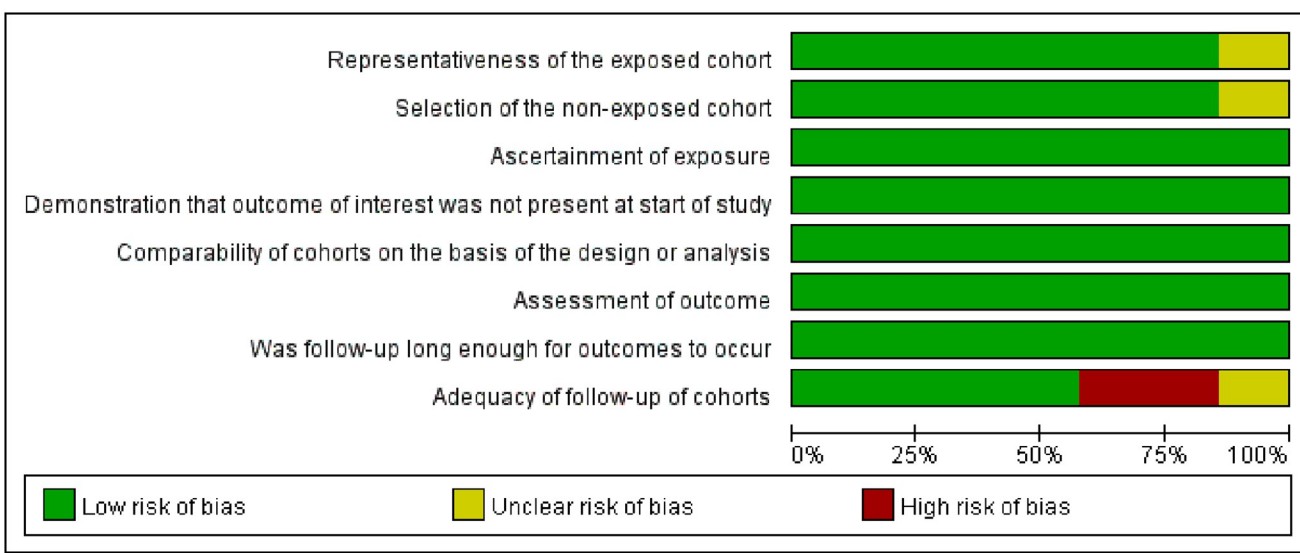

**Fig 2. Quality assessment of the included cohort studies.**

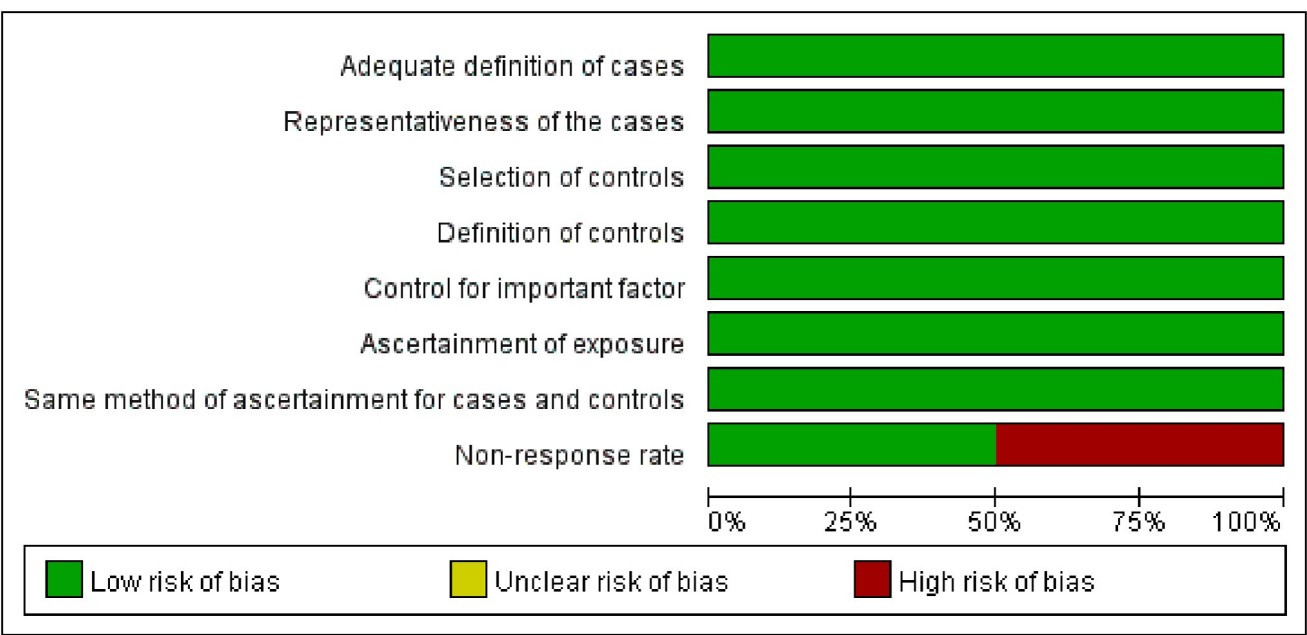

**Fig 3. Quality assessment of the included case-control studies.**

dysmenorrhea, smoking, prior abdominal or pelvic surgery and age. Sort by the size of the OR value. Meta-analyses demonstrated that use opioids (2 studies, OR = 3.91, 95%CI: 1.08–14.24, S1 Fig), dysmenorrhea (3 studies, OR = 2.51, 95%: 1.25–5.05, S2 Fig), smoking (3 studies, OR = 1.55, 95%: 1.18–2.02, S3 Fig), prior abdominal or pelvic surgery (3 studies, OR = 1.46, 95%CI: 1.16–1.83, S4 Fig) and age (3 studies, OR = 1.03, 95%: 1.02–1.04, S5 Fig) were independent associated factors of sarcopenia (Table 2).

**Surgery-related factors.** There were 7 surgery-related factors were assessed in the studies included in the review, including perioperative transfusion, concomitant bowel surgery,

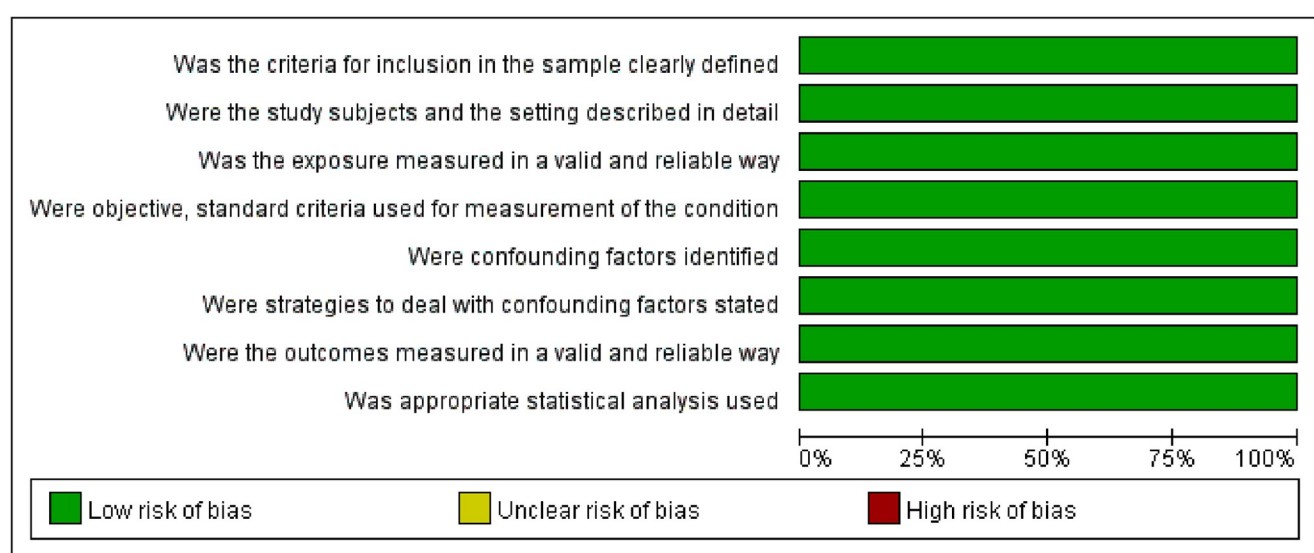

**Fig 4. Quality assessment of the included cross-sectional studies.**

**Table 2. Pooled ORs and 95%CI for risk factors of POI in hysterectomy patients.**

| Risk factors | | Number of studies | Heterogeneity | | OR(95%CI) |
|---|---|---|---|---|---|
| | | | $I^2$ (%) | P | |
| Personal factors and previous history of disease | Use opioids | 2 | 60.8 | < .001 | 3.91 (1.08, 14.24) |
| | Dysmenorrhea | 3 | 55.1 | < .001 | 2.51 (1.25, 5.05) |
| | Smoking | 3 | 14.7 | < .001 | 1.55 (1.18, 2.02) |
| | Prior abdominal or pelvic surgery | 3 | 0 | < .001 | 1.46 (1.16, 1.83) |
| | Age | 3 | 48.3 | < .001 | 1.03 (1.02, 1.04) |
| Surgery-related factors | Perioperative transfusion | 2 | 45.5 | < .001 | 4.50 (3.29, 6.16) |
| | Concomitant bowel surgery | 2 | 32.7 | < .001 | 3.79 (1.86, 7.71) |
| | Anesthesia technique (general anesthesia) | 2 | 0 | < .001 | 2.73 (1.60, 4.66) |
| | Adhesiolysis | 6 | 0 | < .001 | 1.97 (1.52, 2.56) |
| | Duration of operation | 2 | 36.7 | < .001 | 1.78 (1.32, 2.40) |
| | Operation approach (laparoscopic hysterectomy) | 2 | 0 | < .001 | 0.43 (0.29, 0.64) |
| | Operation approach (vaginal hysterectomy) | 2 | 0 | < .001 | 0.35 (0.18, 0.69) |

Notes: OR: odds ratio, CI:.confidence interval.

anesthesia technique (general anesthesia), adhesiolysis, duration of operation, operation approach (laparoscopic hysterectomy) and operation approach (vaginal hysterectomy). Sort by the size of the OR value. Meta-analyses showed that perioperative transfusion (2 studies, OR = 4.50, 95%CI: 3.29–6.16, S6 Fig), concomitant bowel surgery (2 studies, OR = 3.79, 95% CI: 1.86–7.71, S7 Fig), anesthesia technique (2 studies, general anesthesia) (OR = 2.73, 95%CI: 1.60–4.66, S8 Fig), adhesiolysis (6 studies, OR = 1.97, 95%CI: 1.52–2.56, S9 Fig), duration of operation (2 studies, OR = 1.78, 95%CI: 1.32–2.40, S10 Fig), operation approach (laparoscopic hysterectomy) (2 studies, OR = 0.43, 95%CI: 0.29–0.64, S11 Fig) and operation approach (vaginal hysterectomy) (2 studies, OR = 0.35, 95%CI: 0.18–0.69, S11 Fig) were independent associated factors of sarcopenia (Table 2).

**Other evaluated conditions not associated with POI.** Multiple conditions evaluated in our meta-analysis did not reach statistical significance and cannot be considered risk factors for POI. These included BMI, which was assessed in 3 studies and found no significant association between elevated BMI and POI (OR = 1.00, 95%CI: 0.98–1.01, S12 Fig) [7, 17, 18]. Similar findings were found for diabetes (2 studies [7, 18], OR = 1.25, 95%CI: 0.88–1.80, S13 Fig), endometriosis (2 studies [7, 17], OR = 1.20, 95%CI: 0.87–1.66, S14 Fig) and nonwhite race (2 studies [7, 17], OR = 1.35, 95%CI: 0.62–2.95, S15 Fig).

## Study risk-of-bias assessment

Due to the small number of articles included in this paper, publication bias and sensitivity analysis were only performed for risk factors with relatively large number of included articles. As shown in the Fig 5, the funnel plot is symmetrical, so no publication bias is observed. As shown in the Fig 6, the combined results of the remaining studies are still meaningful after the literature is eliminated one by one, indicating that the analysis results are not easy to change significantly due to changes in the number of studies, and have a certain degree of robustness.

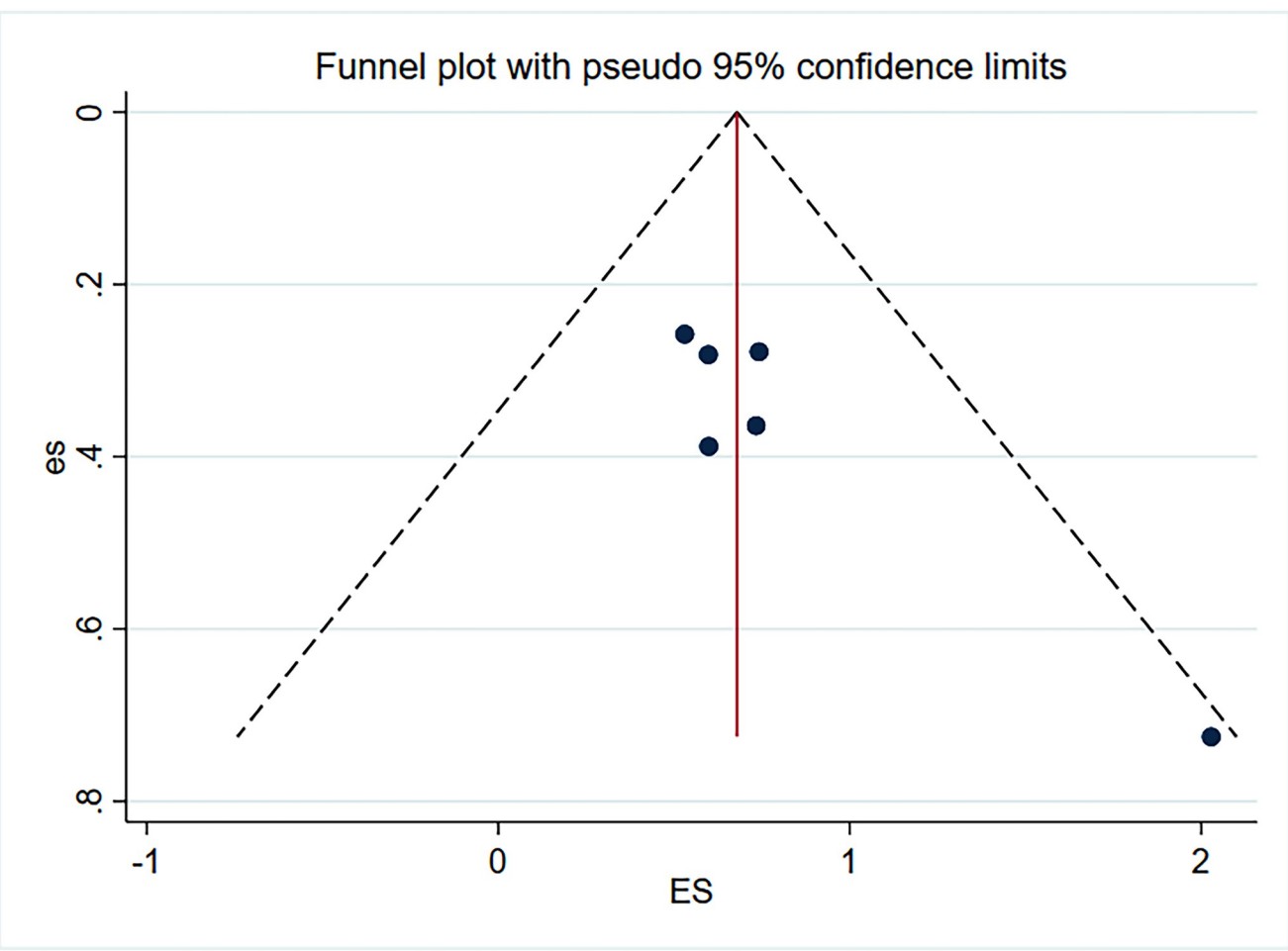

**Fig 5. Funnel plot of adhesiolysis.**

## Discussion

This study comprehensively searched relevant literature at home and abroad, and finally included 11 studies [7, 15–24], including seven retrospective cohort studies, two case-control studies and two cross-sectional studies, which showed that the incidence of POI in hysterectomy patients ranged from 0.5% to 33.3%. The factors influencing the occurrence of POI in different studies were different. A meta-analysis of the included literature found that use opioids, dysmenorrhea, smoking, prior abdominal or pelvic surgery, age, perioperative transfusion, concomitant bowel surgery, anesthesia technique (general anesthesia), adhesiolysis and duration of operation were risk factors for the occurrence of POI in hysterectomy patients. Operation approach (laparoscopic hysterectomy) and operation approach (vaginal hysterectomy) are protective factors of POI. BMI, diabetes, endometriosis and nonwhite race were not associated factors for POI. The heterogeneity of most variables in the included studies was small, and sensitivity analysis showed that most of the meta-analysis results were stable. Consistency analysis was conducted on baseline data such as age and gender of the subjects in the included literatures, indicating that the two groups were comparable ($P > 0.05$), and the results were of certain reliability.

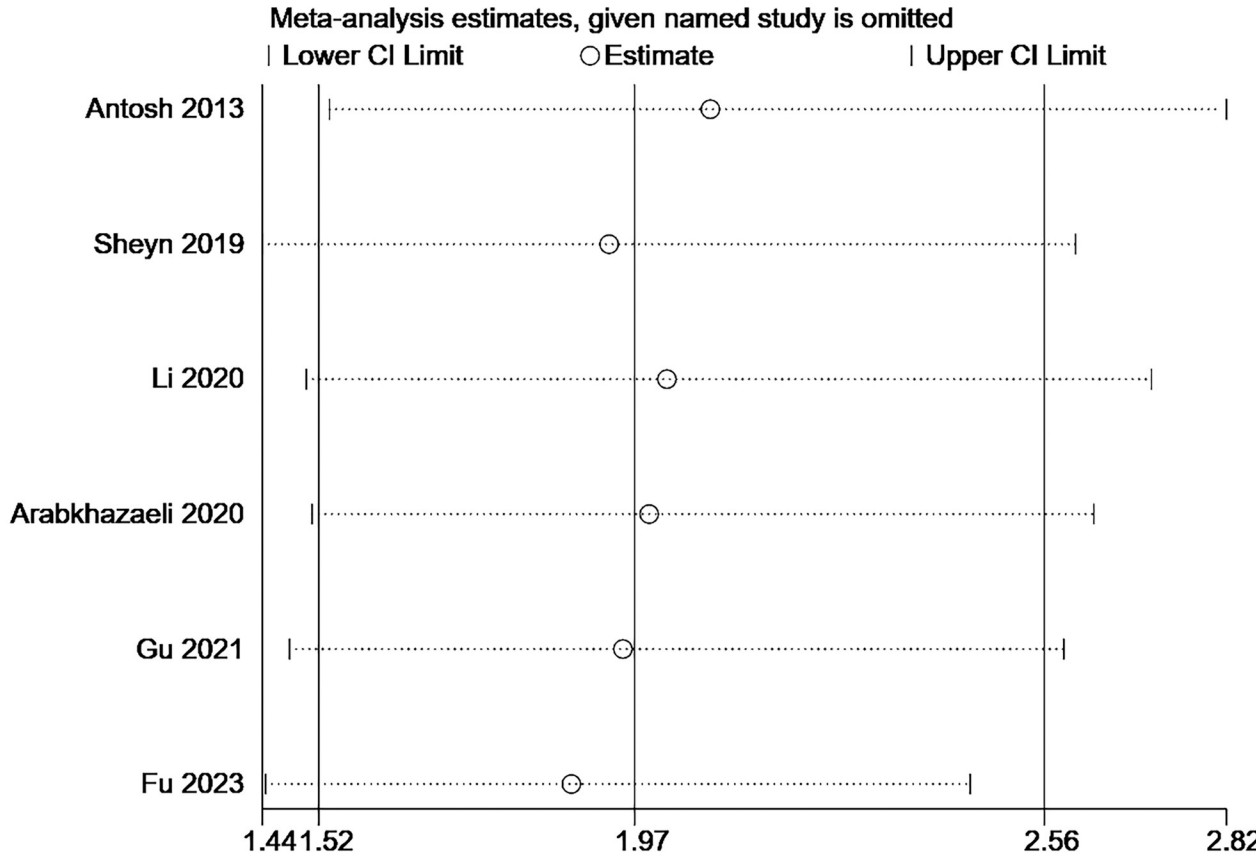

**Fig 6. Sensitivity analysis of studies.**

### Personal factors and previous history of disease

In terms of personal factors and previous history of disease, use opioids, dysmenorrhea, smoking, prior abdominal or pelvic surgery and age were strongly associated with the development of POI. Opioids are commonly used as analgesics during surgery [25]. The activation of opioid receptors in the body under surgical stimulation, combined with the intraoperative application of opioid labor pains, inhibited mesenteric nerve plexus, blocked acetylcholine release, increased intestinal muscle tension, significantly decreased gastrointestinal peristalsis function, and easily induced POI [26]. This is consistent with the findings of Quiroga-Centeno et al. [27], which suggest that opioids are detrimental to gastrointestinal recovery, whether used after intestinal surgery or hysterectomy. Most patients with dysmenorrhea suffer from diseases such as endometriosis, and the fibroid tissue needs to be removed before surgery [28]. Moreover, dysmenorrhea is closely related to changes in central nervous system function, and more analgesic drugs may be required during perioperative period [29], which will inhibit gastrointestinal function and increase the risk of POI. In addition, tobacco contains nicotine substances, which can stimulate capillaries and cause contraction, affecting the blood supply at the surgical site, resulting in postoperative incision infection and poor healing, and thus affecting the postoperative gastrointestinal function recovery of patients [30]. Cribb et al. [31] also support this view. The reason why this factor has low heterogeneity is that smoking is divided into former smokers (who have quit smoking) and current smokers, and the methods of data collection were inconsistent among different studies. The risk of POI was also higher in patients

who had prior abdominal or pelvic surgery. The structure of the abdominal organs has changed after the previous abdominal surgery, the intestinal irritation will increase during the next operation [32]. Fang et al. [33] also believe that the structural changes of abdominal organs increase the risk of POI. While, patients with a history of abdominal surgery may have abdominal adhesion, which increases the risk of gastrointestinal surgical complications [34]. With the increase of age, the gastrointestinal peristalsis function of the elderly is worse than that of the young, and the decline of immunity leads to the intestinal tract is more susceptible to infection, which affects the gastrointestinal motility [35]. However, this risk factor has low heterogeneity, which may be due to differences in the average age of the included studies.

## Surgery-related factors

In terms of surgery-related factors, perioperative transfusion, concomitant bowel surgery, anesthesia technique (general anesthesia), adhesiolysis and duration of operation were strongly associated with the development of POI. Operation approach (laparoscopic hysterectomy) and operation approach (vaginal hysterectomy) were the protective factors of POI. Perioperative transfusion is also an important risk factor for POI. However, this risk factor has low heterogeneity, which may be due to the fact that the amount of fluid injected in each study was not defined and the cutoff value was not calculated. Perioperative transfusion is mainly used to correct anemia, and POI may be related to the cause of the transfusion (such as high blood loss) rather than the transfusion itself [36], and the large amount of intraoperative blood loss may also indicate the severity and complexity of the patient's condition. The risk of POI was higher in patients who were concomitant bowel surgery than in patients undergoing hysterectomy alone. Surgical instruments will irritate the intestine and mesentery, which is likely to damage local vessels, nerves and the intestinal wall, resulting in damaged blood circulation, leading to temporary loss or decline of intestinal function [37]. Patients undergoing surgery under general anesthesia are more likely to develop POI compared to local anesthesia. General anesthesia drugs are injected intravenously or intramuscularly into the body, resulting in temporary suppression of the central nervous system, which can inhibit gastrointestinal peristalsis, leading to bloating and even POI [38, 39]. Holte et al. [40] have shown that epidural block anesthesia is more beneficial for gastrointestinal recovery. Moreover, patients who underwent adhesiolysis were more likely to develop POI. Adhesiolysis is an additional intestinal operation. The reason for adhesion is that in the process of abdominal operation, the body is easy to be infected by various bacteria or germs, and the emergence of inflammatory reaction often leads to the infiltration of a large amount of fibrin liquid in the body, forming a lot of adhesive areas and lead to POI [41]. Al-Sunaidi et al. [20] and Montz et al. [24] have the same point of view. With the prolongation of operation time, the use of anesthetic drugs will also increase, so that more acetylcholine is released, which has an inhibitory effect on intestinal smooth muscle and thus affects intestinal peristalsis [26]. However, this risk factor has low heterogeneity, which may be due to the different concept of long time in each article. Compared with total abdominal hysterectomy, laparoscopic and transvaginal hysterectomy were protective factors for POI. These two surgical methods have the advantages of less trauma, quick recovery after surgery, little impact on the abdominal and pelvic organs, and less interference to the intestine [42]. In addition, laparoscopic hysterectomy can safely perform adhesion decomposition, and the probability of postoperative uterine adhesion is low [43]. This view is echoed in other studies included in this study. Schindlbeck et al. [21] also found that laparoscopic surgery is a safer and less invasive method, and recovery after surgery is faster. Although vaginal hysterectomy is faster and the results are similar, laparoscopy is more advantageous in terms of observing intra-abdominal location and pathology [21]. Al-Sunaidi et al. [20] concluded that total

abdominal hysterectomy was the most common cause of small intestinal obstruction among all gynecological benign operations, and small intestinal obstruction rarely occurred in laparoscopic operations. However, there are some different voices. Muffly et al. [23] concluded that the route of hysterectomy did not seem to affect the risk of SBO. Surgical route as a risk factor is still controversial and needs to be further demonstrated.

The included studies still have risk factors to prove. Arabkhazael et al. [18] suggested that patients with malignant uterine pathology were more likely to develop POI. POI was more likely to occur in patients undergoing hysterectomy with concomitant radiotherapy [24]. However, limited by the number of studies and the data in the original studies, these important risk factors could not be analyzed.

Based on the results of our article, it is suggested that medical staff should pay more attention to patients with dysmenorrhea, advanced age, smoking and history of abdominal and pelvic surgery. Doctors should carefully select operation approach, anesthesia technique and analgesic drugs according to the specific conditions of patients, so as to minimize perioperative transfusion and adhesion, shorten duration of operation and reduce intraoperative trauma. In addition, nurses should provide a detailed preoperative education, smokers should quit smoking, and the nurse should ask the patient's families to supervise. Patients should not eat too much high-fat, high-sugar food before surgery, and should eat more fiber-rich food and water. As well as, the causes of dysmenorrhea should be carefully investigated before operation, and pelvic surgery should be chosen carefully. Minimally invasive surgical methods should be chosen as much as possible. Before surgery, surgeons and nurses should be fully familiar with surgical operations and procedures and make adequate preoperative preparations. During the operation, the nurse should closely cooperate with the doctor and try to control the operation time. General anesthesia should be avoided. Some authors have proved that epidural anesthesia is beneficial to the recovery of POI [40]. After operation, multi-mode analgesia should be taken, the use of opioids should be reduced. Opioids should be replaced with nonsteroidal anti-inflammatory drugs, epidural analgesia, etc. Besides, medical staff can encourage and assist patients to get out of bed early, resume oral eating. It is recommended that patients chew gum, and oral laxatives to promote the rapid recovery of gastrointestinal function.

## Study strengths and limitations

Firstly, as far as we know, so far, this is the first systematic review and meta-analysis to focus on factors associated with POI in hysterectomy. Secondly, we conducted an extensive literature search, thoroughly screening research papers, as well as other relevant literature, to minimize the possibility of missing any research. Thirdly, most of the included studies were of high quality with reliable results.

However, our view also had some limitations. First of all, there are relatively few studies on the risk factors of POI in hysterectomy patients, so the number of literatures included in the meta-analysis was too small to conduct subgroup analysis for risk factors with large heterogeneity. Therefore, it is suggested to carry out a large number of multi-center, large-sample and high-quality studies to unify the diagnostic criteria of POI and further clarify the risk factors, so as to provide practical and reliable basis for clinical medical staff to manage POI and reduce the incidence of POI. Secondly, due to the limitation of the original literature data, the risk factors closely related to POI, such as uterine pathology and concomitant radiotherapy could not be analyzed in this study. Thirdly, the literatures included in this study were observational studies, which could not determine the exact causal relationship between the POI and its risk factors.

## Conclusions

In summary, this study found that use opioids, dysmenorrhea, smoking, prior abdominal or pelvic surgery, age, perioperative transfusion, concomitant bowel surgery, anesthesia technique (general anesthesia), adhesiolysis, duration of operation, Operation approach (laparoscopic hysterectomy) and operation approach (vaginal hysterectomy) were influenced factors for POI in patients undergoing hysterectomy. This suggests that medical staff should fully assess the patient's condition, reduce the stimulation of the patient's intestinal wall during the operation, and nurses and physicians should closely cooperate to control the operation time as much as possible. After the operation, encourage and assist patients to get out of bed early and promote the recovery of gastrointestinal function. Our findings provide a basis for screening high-risk groups for POI, provide a reference for perioperative decision-making, and help reduce the incidence of POI and the resulting medical and family burden. However, further studies are needed to further elucidate the unexplored risk factors for POI, such as BMI, diabetes, endometriosis, uterine pathology, concomitant radiotherapy and so on.

## Supporting information

**S1 Checklist. PRISMA 2020 checklist.**
(DOCX)

**S1 File. Forest plots of factors.**
(DOCX)

**S1 Table. Search strategies in each database.**
(DOCX)

## Acknowledgments

We want to acknowledge the Zhejiang Chinese Medical University for its continuous support and companionship during the students' research.

## Author Contributions

**Conceptualization:** Zhuoer Hou, Ting Liu.

**Data curation:** Zhuoer Hou, Xiaoyan Li, Hangpeng Lv.

**Formal analysis:** Zhuoer Hou, Xiaoyan Li.

**Investigation:** Hangpeng Lv.

**Supervision:** Qiuhua Sun.

**Writing – original draft:** Zhuoer Hou.

**Writing – review & editing:** Ting Liu, Qiuhua Sun.

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
