## [Decision Letter · Decision Letter 0]

5 Jul 2024

PONE-D-24-11634Risk factors for postoperative ileus in hysterectomy: A Systematic Review and Meta-analysisPLOS ONE

Dear Dr. Sun,

Thank you for submitting your manuscript to PLOS ONE. After careful consideration, we feel that it has merit but does not fully meet PLOS ONE’s publication criteria as it currently stands. Therefore, we invite you to submit a revised version of the manuscript that addresses the points raised during the review process. You need to take the advices of reviewer 1 and answer the questions as well if possible.

We look forward to receiving your revised manuscript.

Kind regards,

Mert Tanal

Academic Editor

PLOS ONE

Journal Requirements:

"This project is jointly supported by The National Natural Science Foundation of China (No. 8197152282)."

Reviewers' comments:

Reviewer's Responses to Questions

**Comments to the Author**

1. Is the manuscript technically sound, and do the data support the conclusions?

Reviewer #1: Yes

2. Has the statistical analysis been performed appropriately and rigorously? 

Reviewer #1: Yes

3. Have the authors made all data underlying the findings in their manuscript fully available?

Reviewer #1: No

4. Is the manuscript presented in an intelligible fashion and written in standard English?

Reviewer #1: Yes

5. Review Comments to the Author

Reviewer #1: 1) The objective was to assess risk factors for POI , how then was preventive measures and interventions arrived at?

2) In terms of literature search and studies included in the review, what does the statement from inception to march 2024 mean?

3) The tense from line 74 to 78 mean need to be corrected under search strategy

4) The discussion section needs to bring out what is in other literature with comparison and contrast of the findings with the review

6. PLOS authors have the option to publish the peer review history of their article (what does this mean?). If published, this will include your full peer review and any attached files.

Reviewer #1: No

---

## [Author Response · Author response to Decision Letter 0]

17 Jul 2024

Submission ID: PONE-D-24-11634

Title: Risk factors for postoperative ileus in hysterectomy: A Systematic Review and Meta-analysis

Correspondence Author: Qiuhua Sun

Dear Editor,

Thank you for giving us an opportunity to revise and resubmit our manuscript. And we appreciated the constructive criticisms and valuables suggestions of you and the reviewers. All your comments are helpful for improving not only our submission but also the further research work.

All of us have seriously read and discussed the comments. And we have carefully revised the resubmitted 'Manuscript' according to the reviewer's comments, and we have submitted the 'Revised Manuscript with Track Changes' with specific revised marks in the supplementary materials. In addition, we have made linguistic improvements in the spelling and readability of the article, and we have invited native English speakers to touch up our manuscript. We sincerely hope that as a result of this revision, the language level of the manuscript has been substantially improved. 

We have responded to each of the reviewer's comments in the text and marked the responses in blue. We hope that you will be satisfied with our revision. If there are still some more requirement, please let us know. And we have been preparing to do our best.

Best regards!

Qiuhua Sun 

The College of Nursing, Zhejiang Chinese Medical University

No.584, Binwen Road, Hangzhou, Zhejiang Province 310053

China

Email: sunqiuhua@zcmu.edu.cn

Response to Journal Requirements:

1.Please ensure that your manuscript meets PLOS ONE's style requirements, including those for file naming. The PLOS ONE style templates can be found at https://journals.plos.org/plosone/s/file?id=wjVg/PLOSOne_formatting_sample_main_body. Pdf 

and https://journals.plos.org/plosone/s/file?id=ba62/PLOSOne_formatting_sample_title_authors_affiliations.pdf.

Thank you for your advice. We have made changes to the format of the article according to PLOS ONE's style requirements, including font size of the title, naming of tables and figures, file naming, formatting of references, etc.

2. Thank you for stating the following financial disclosure: "This project is jointly supported by The National Natural Science Foundation of China (No. 8197152282)." Please state what role the funders took in the study.  If the funders had no role, please state: "The funders had no role in study design, data collection and analysis, decision to publish, or preparation of the manuscript." If this statement is not correct you must amend it as needed. Please include this amended Role of Funder statement in your cover letter; we will change the online submission form on your behalf.

Thank you for your advice. The funders had no role in study design, data collection and analysis, decision to publish, or preparation of the manuscript. We have stated this in our cover letter.

Thank you for your suggestions. All data are in the manuscript and/or supporting information files. We have state this at the end of the manuscript.

Thank you for your suggestions. We have added the captions of the Supporting Information files at the end of your manuscript and updated any in-text citations to match accordingly.

5.Please review your reference list to ensure that it is complete and correct. If you have cited papers that have been retracted, please include the rationale for doing so in the manuscript text, or remove these references and replace them with relevant current references. Any changes to the reference list should be mentioned in the rebuttal letter that accompanies your revised manuscript. If you need to cite a retracted article, indicate the article’s retracted status in the References list and also include a citation and full reference for the retraction notice.

Thank you for your advice. We have checked the references and found no retraction of the cited papers. But we have changed the format of the references.

Response to Reviewers' comments (Reviewer #1):

1.Have the authors made all data underlying the findings in their manuscript fully available?

Reviewer #1: No

Thank you very much for your comments. All data are in the manuscript and/or supporting information files. We have state this at the end of the manuscript.

2.The objective was to assess risk factors for POI, how then was preventive measures and interventions arrived at?

Thank you very much for your kind suggestions, which will help us to improve the structure of the paper. Based on the results of our study and reviewing the relevant literature, we added the content of preventive measures and interventions for POI after hysterectomy accordingly around three aspects: preoperative (education, diet, smoking cessation, etc.), intraoperative (controlling the duration of the operation, avoiding general anaesthesia, etc.), and postoperative (getting out of bed at an early stage, decreasing the use of opioids for analgesia, chewing gum, etc.) (Line 340-357).

3. In terms of literature search and studies included in the review, what does the statement from inception to march 2024 mean?

Thank you for your detailed review of our study, your suggestions have been instrumental in our revisions. This may be an incorrect English expression. We replaced that with "The search period was restricted from the earliest records to March 2024." (Lines 77-78)

4. The tense from line 74 to 78 mean need to be corrected under search strategy.

Your point about the tense is crucial, and we have made corresponding adjustments. There are some inconsistencies in the tenses and expressions of the methods in the manuscript. We studied the journal style of PLOS ONE and the methods of similar studies in authoritative journals (references [1-3] are appended later). We modified the language of my manuscript by using the past tense and more professional expressions, hoping to improve the readability of this part (Lines 78-91). Thank you for your careful review and practical suggestions.

[1] Li X, He J, Sun Q. Sleep Duration and Sarcopenia: An Updated Systematic Review and Meta-Analysis[J]. Journal of the American Medical Directors Association, 2023, 24(8): 1193-1206.e5.

[2] Gowrishankar S, Smith M E, Creber N, et al. Immunosuppression in stem cell clinical trials of neural and retinal cell types: A systematic review[J]. PLoS ONE, 2024, 19(7): e0304073.

[3] Kassaw A, Asferie W N, Azmeraw M, et al. Incidence and predictors of tuberculosis among HIV-infected children after initiation of antiretroviral therapy in Ethiopia: A systematic review and meta-analysis[J]. PLoS ONE, 2024, 19(7): e0306651.

5. The discussion section needs to bring out what is in other literature with comparison and contrast of the findings with the review.

We value your suggestions in the discussion and have made adjustments accordingly. In the discussion, We chose the factors of use opioids (Lines 274-276), smoking (Lines 383), prior abdominal or pelvic surgery (Lines 288-289), anesthesia technique (Lines 312-313), duration of operation (Lines 317-320), operation approach (laparoscopic hysterectomy) (Lines 326-328), and operation approach (vaginal hysterectomy) (Lines 330-333) to be discussed in comparison with other literature contents. These newly added literature and comparisons are indeed helpful to understand the results of this study, and we look forward to your feedback.

---

## [Editor Report · Decision Letter 1]

18 Jul 2024

Risk factors for postoperative ileus in hysterectomy: A Systematic Review and Meta-analysis

PONE-D-24-11634R1

Dear Dr. Qiuhua Sun,

We’re pleased to inform you that your manuscript has been judged scientifically suitable for publication and will be formally accepted for publication once it meets all outstanding technical requirements.

Kind regards,

Mert Tanal

Academic Editor

PLOS ONE
---

## [Editor Report · Acceptance letter]

22 Jul 2024

PONE-D-24-11634R1 

PLOS ONE

Dear Dr. Sun, 

I'm pleased to inform you that your manuscript has been deemed suitable for publication in PLOS ONE. Congratulations! Your manuscript is now being handed over to our production team.

Kind regards, 

on behalf of

Dr. Mert Tanal 

Academic Editor

PLOS ONE